# Synthesis of Chitosan Oligosaccharide-Loaded Glycyrrhetinic Acid Functionalized Mesoporous Silica Nanoparticles and In Vitro Verification of the Treatment of APAP-Induced Liver Injury

**DOI:** 10.3390/molecules28104147

**Published:** 2023-05-17

**Authors:** Xinghua Guo, Chengcheng Zhang, Yan Bai, Qishi Che, Hua Cao, Jiao Guo, Zhengquan Su

**Affiliations:** 1Guangdong Engineering Research Center of Natural Products and New Drugs, Guangdong Pharmaceutical University, Guangzhou 510006, China; 2Guangdong Metabolic Disease Research Center of Integrated Chinese and Western Medicine, Guangdong Pharmaceutical University, Guangzhou 510006, China; 3School of Public Health, Guangdong Pharmaceutical University, Guangzhou 510310, China; 4Guangzhou Rainhome Pharm & Tech Co., Ltd., Science City, Guangzhou 510663, China; 5School of Chemistry and Chemical Engineering, Guangdong Pharmaceutical University, Zhongshan 528458, China

**Keywords:** acetaminophen, chitosan oligosaccharide, drug-induced liver injury, glycyrrhetinic acid modification, mesoporous silica

## Abstract

Objective: the study was to find a suitable treatment for acute drug-induced liver injury. The use of nanocarriers can improve the therapeutic effect of natural drugs by targeting hepatocytes and higher loads. Methods: firstly, uniformly dispersed three-dimensional dendritic mesoporous silica nanospheres (MSNs) were synthesized. Glycyrrhetinic acid (GA) was covalently modified on MSN surfaces through amide bond and then loaded with COSM to form drug-loaded nanoparticles (COSM@MSN-NH_2_-GA). The constructed drug-loaded nano-delivery system was determined by characterization analysis. Finally, the effect of nano-drug particles on cell viability was evaluated and the cell uptake in vitro was observed. Results: GA was successfully modified to obtain the spherical nano-carrier MSN-NH_2_-GA (≤200 nm). The neutral surface charge improves its biocompatibility. MSN-NH_2_-GA has high drug loading (28.36% ± 1.00) because of its suitable specific surface area and pore volume. In vitro cell experiments showed that COSM@MSN-NH_2_-GA significantly enhanced the uptake of liver cells (LO2) and decreased the AST and ALT indexes. Conclusion: this study demonstrated for the first time that formulation and delivery schemes using natural drug COSM and nanocarrier MSN have a protective effect on APAP-induced hepatocyte injury. This result provides a potential nano-delivery scheme for the targeted therapy of acute drug-induced liver injury.

## 1. Introduction

Acetaminophen (APAP), which is also known as N-acetyl-p-aminophenol or paracetamol, is among the most frequently used tablets due to its analgesic and antipyretic properties [1,2]. Especially in the era of the COVID-19 virus epidemic, because of its low price and significant effect on fighting fever and muscle soreness, it is often combined with other drugs and is easily overused frequently [3,4,5]. Therefore, the safety of APAP is questionable. Studies have shown that its overdose might lead to hepatotoxicity and acute liver failure (ALF) [6,7]. APAP mediates the formation of N-acetyl-p-benzoquinone imine through cytochrome P450 in the human body, leading to mitochondrial oxidative stress and activation of c-Jun N-terminal kinase, which will lead to nuclear translocation of mitochondrial proteins and induce DNA fragmentation, eventually leading to liver cell necrosis [8]. The drug-induced liver damage (DILI) observed with APAP is the second main reason for liver transplantation in humans [9]. APAP-induced DILI is responsible for 50% of cases of acute liver failure in the United States and Europe [10], whereas in China, APAP-induced DILI is responsible for 50.8% of all cases of anti-inflammatory- and analgesic-induced DILI [11]; therefore, it is urgent to develop safer and more effective drugs to treat liver injury induced by APAP. At present, N-acetylcysteine (NAC) is the only clinically approved antidote for APAP-induced liver damage [12,13,14]. Because of its poor bioavailability, complicated drugs, limited administration time, and many side effects, it is necessary to design and construct a new nano-drug delivery system to find a solution [15,16,17,18].

Mesoporous silicon nanocarriers (MSNs) have high drug loading capacity and can load small molecular drugs, especially hydrophobic drugs. Up to now, MSNs have been developed for targeted drug, gene and protein delivery, and composite nano-drugs for diagnostic biological imaging, tissue engineering, cancer treatment, vaccine development, biomaterials, and diagnosis and treatment [19,20,21]. At the same time, nano-loaded drugs can avoid drug degradation and physiological toxicity to healthy tissues caused by the premature exposure of drugs [22,23]. Mesoporous silica with various specific modifications has attracted people’s interest as the carrier of nanoparticle drug delivery system. This approach enhances cellular-specific uptake to increase intracellular drug concentration and retain the drug in the targeted tissue [24,25]. In addition, targeted nanoparticles include excessive drug load and ensure targeted delivery and improve bioavailability [19].

Glycyrrhetinic acid (GA) is an active aglycone of glycyrrhizic acid. Studies have proven that GA has numerous advisable pharmacological activities, such as anti-inflammatory, antiviral and antiulcer activities [26,27]. Because GA molecules can provide a hydrophobic moiety and bind liver-targeting ligands, GA-mediated drug carrier structures have emerged as a novel liver-targeting platform [28,29]. Studies have confirmed that GA receptors are present on the surface of liver (parenchymal) cell membranes, and GA can be highly accumulated in the liver [30]. Studies have confirmed that GA receptors are present on the surface of liver (parenchymal) cell membranes, and GA can be highly accumulated in the liver. Compared to carriers not modified with GA, those modified with GA were reported to be more effective in livers or in targeting livers [30]. In addition, GA-modified nanoparticles may exhibit an ability to distinguish normal liver tissue from diseased or damaged liver tissue, resulting in higher therapeutic efficacy and safety [31]. In a word, it is a very promising scheme to embed GA into MSN and load drugs to treat diseases in the liver.

COSM, with an average molecular weight of less than 3000, is a natural product extracted from shrimp and crab shells that exhibits antioxidant, anti-inflammatory and other biological activities [32,33,34]. Our research group has performed research on chitosan and chitosan oligosaccharide regarding weight loss and liver protection for a long time. In particular, the team has a good research foundation on the antioxidant and anti-inflammatory activities and mechanisms of chitosan oligosaccharide on the liver. Previous studies have found that COSM has a good protective effect on liver injury [35,36,37,38,39]. In this study, GA was selected as the target ligand to be covalently modified on the structure of MSNs. The ensuing glycyrrhetinic acid-functionalized mesoporous silica nanoparticles (MSN-NH_2_-GA) have potential for transferring unique capsules to LO2 cells. COSM, a polysaccharide product with anti-inflammatory and antioxidant effects, was loaded into MSN-NH_2_-GA. The covalent bond between GA and MSN was investigated by Fourier transform infrared spectroscopy (FTIR). The structures and morphologies of MSNs and MSN-NH_2_-GA were investigated via transmission electron microscopy (TEM) and dynamic light scattering (DLS). LO2 cells were selected to explore the cytotoxicity and cell uptake by focusing on the effectiveness of COSM-loaded MSN-NH_2_-GA.

## 2. Results and Discussions

### 2.1. Synthesis and Characterization of MSN-NH_2_-GA and COSM@MSN-NH_2_-GA

The nanomedicine, labelled COSM@MSN-NH_2_-GA, was synthesized in several steps (Figure 1). Briefly, uniformly dispersed three-dimensional dendritic mesoporous silica nanospheres were synthesized by referring to the preparation method by Shen’s team [40]. These 3D dendritic MSNs exhibit unique advantages in protein loading and release due to their adjustable large porosity and intelligent layered mesostructure. More importantly, the release rate depends in part on graded biodegradation as 3D dendritic MSNs with larger pore sizes have a faster rate of biological interpretation [41,42]. In this study, GA was selected as the target ligand, and by grafting alkoxysilane, MSNs were externally functionalized and finally anchored by amino reaction with amino-modified GA [43]. COSM was loaded in ethanol by impregnation [44]. The formed glycyrrhetinic acid-functionalized MSN nanoparticles exhibit the potential to specifically deliver drugs to hepatocytes [42].

As shown in Figure 1a,b, SEM revealed that the MSN nanoparticles are spherical and exhibit a uniform particle size distribution. TEM showed that the MSN nanoparticles are spherical with clear and uniform mesoporous channels on the surface and a uniform particle size, which is consistent with the SEM results. After targeted modification, MSN-NH_2_-GA nanoparticles with a uniform particle size and complete morphology were obtained. The particle size distributions of MSN and MSN-NH_2_-GA nanoparticles are (156.7 ± 61.7) nm and (190.7 ± 78.1) nm, respectively (Figure 1c). The particle size of the nanoparticles gradually increases with the modification process. The nanoparticles are dispersed in different solvents, and the measured particle size is also inconsistent because the DLS measurement conditions are in aqueous solution. Therefore, the DLS results are slightly larger than the TEM particle size results. The zeta potential of the nanoparticles also reflects the macroscopic changes in the surface modification of the nanoparticles. Figure 1d shows that the zeta potential of the blank MSN was −40.42 ± 22.11 mV due to the presence of silanol groups on the MSN surface and its negative charge. After amino modification occurred, the amino group (positively charged) replaced silanol (which is negatively charged), so the zeta potential changed from negative to positive, and the zeta potential of MSN-NH_2_ was 58.62 ± 4.479 mV. When glycyrrhetinic acid was docked, part of the amino group was consumed and covered so that the positive charge decreased, and the zeta potential of MSN-NH_2_-GA became (7.013 ± 4.132) mV.

To decide the chemical grafting of exceptional practical groups, MSN, MSN-NH_2_ and MSN-NH_2_-GA were characterised using distinct methods and after every reaction step. Using FTIR spectroscopy, we can see the functionalization manner of nanoparticles. Figure 2 shows that all samples show the skeleton absorption peaks of silicon-based materials, namely, the Si-O-Si symmetrical stretching vibration peak (800 cm^−1^), the Si-O-Si asymmetric stretching vibration peak (1085 cm^−1^), and the Si-OH stretching vibration peak (960 cm^−1^). The infrared spectra peaks of MSN-NH_2_ are located at 2922 cm^−1^ and 2855 cm^−1^, which are the C-H stretching vibrations of APTES, indicating that the amino group was correctly modified. The amide II band at 1445 cm^−1^ and the C=C stretching vibration of GA at 1545 cm^−1^ indicate that the GA molecule is modified to MSN by amide bonds.

As seen in Figure 3a,b, the adsorption isotherm conforms to the Langmuir IV isotherm, and the nanomaterial exhibits a mesoporous structure. It can be seen that, when the relative pressure P/P_0_ < 0.35, N_2_ is present on the surface of the material channel. With single molecule and multimolecular layer adsorption, the amount of adsorption slowly increases. Under a relative pressure of 0.35 < P/P_0_ < 0.8, an obvious capillary condensation step can be observed, and the adsorption amount increases, indicating that the pore size distribution is wide. When 0.8 < P/P_0_ < 0.9, the nitrogen adsorption and the outer surface curve are gentle. When P/P_0_ > 0.9, there is a hysteresis loop. At this time, nitrogen adsorption occurs in the gap between the particles, the adsorption capacity increases, and the curve shows an additional large jump. The postgrafting method was used to modify the targeting group so that group modification would also occur in the pores of the nanoparticles. Figure 3b shows that the modification of amino and glycyrrhetinic acids exhibited a certain covering effect on the nanoparticles. As seen in Figure 4a,b, the MSNs had specific surfaces of 565.27 m^2^/g, contained pores with a size of 6.15 nm, and a volume of 1.18 cm^3^/g. MSN-NH_2_-GA had specific surface areas of 245.83 m^2^/g, showed pores with a size of 6.04 nm, and a volume of 0.69 cm^3^/g. Therefore, the specific surface area, pore volume and pore size of MSN-NH_2_-GA decreased accordingly.

To identify the COSM API, nanocarriers (MSN-NH_2_-GA), physically mixed groups, and nanopharmaceutical groups, differential scanning calorimetry (DSC) was performed. The results are shown in Figure 5. The black (A), red (B), blue (C), and green (D) curves represent the COSM, nanocarrier (MSN-NH_2_-GA), physical mixing, and nanomedicine (COSM@MSN-NH_2_-GA) groups, respectively. COSM exhibits an obvious single-melting endothermic peak at approximately 200 °C, while the nanocarrier (MSN-NH_2_-GA) group shows an obvious dehydration peak at approximately 100 °C. The DSC analysis of the physically mixed group contains characteristic absorption peaks of the carriers and COSM, which was because the drug and the carriers were simply mixed. In the DSC analysis of the nanodrug group, the characteristic absorption peak of the main drug (COSM) disappeared, indicating that COSM was present in an amorphous form in the nanodrug group and was no longer present in a crystalline state. These observations indicate that COSM was successfully incorporated into the mesoporous channels of MSN-NH_2_-GA. 

According to the previous experiments, the feasibility of using 3,5-dinitrosalicylic acid (DNS) as a method for the determination of COSM content was determined, and the specific determination conditions were finally optimized and screened out [45,46]. In the drug loading experiment, the dosage ratio of COSM and MSN-NH2-GA is 1:2, and the drug loading time is 12 h, which is the best drug loading condition. The encapsulation efficiency (*EE%*) was 28.36% and the load capacity (*LC%*) was 56.72%.

### 2.2. In Vitro Biological Evaluation

LO2 was cultivated, 8000–10,000 cells per well in a 96-well plate were seeded, the cell state with a microscope was observed after about 10–12 h of culture; APAP modelling groups were set at 0, 2, 4, 6, 8, 10, 12, 14, 16 mM. APAP was given to each group according to the preset setting, and the culture was continued for 3, 6, 12, and 24 h; the cell survival rate was measured by CCK-8 method, and the optimal concentration and time for modelling were determined. According to the results of nanomedicine cytotoxicity, the dosage of COSM@MSN-NH_2_-GA group was low dose (200 μg/mL), medium dose (400 μg/mL), high dose (800 μg/mL); in total, there was a group of six duplicate holes. The drug loading of COSM@MSN-NH2-GA is 28.36% ± 1.00%. By equivalent conversion, the dosage of the free drug COSM was low dose (56 μg/mL), medium dose (113 μg/mL), high dose (226 μg/mL); in total, eight duplicate wells were in each group, and they continued to culture for 12 h after adding the COSM drug. The experiment and results of this part are shown in the annex.

LO2 hepatocytes were treated with APAP for 12 h and then treated with COSM, MSN-NH_2_-GA, and COSM@MSN-NH_2_-GA for 12 h. Then, the growth state of the cells were observed under the microscope. As shown in Figure 6, in the blank nanocarrier groups, MSN-NH_2_-GA(L), MSN-NH_2_-GA(M), and MSN-NH_2_-GA(H), basically no difference in the cell state and the number of dead cells was observed. This indicated that the blank nanocarrier did not improve the decrease of cell viability caused by APAP at each dose. Compared with the MOD group, the cell viability was not affected, which preliminarily proved that the nanoparticles were a safe and non-toxic nanocarrier. In the free drug group, the cell state of the COSM (L) group was close to that of the APAP model group, the normal hepatocytes in the COSM (M) and COSM (H) groups increased, the cell morphology was more normal, and the number of dead cells decreased. Each group treated with the nanodrug showed a decrease in the number of dead cells, but the middle-dose group and the high-dose group displayed the most pronounced effects of cell treatment due to the more normalized cell morphology and lower number of dead cells in these groups. The model group generally showed a substantially higher number of dead cells and significantly fewer normal liver cells than the normal control group. In addition, the appearance of many dead cells changed, displaying abnormal forms and decreased cell activity.

The contents of ALT and AST in the medium were determined by collecting the culture medium, and the therapeutic effect of each administration group on APAP-induced LO2 hepatocyte injury was detected by CCK-8 method. As shown in Table 1, COSM@MSN-NH_2_-GA (M) and COSM@MSN-NH_2_-GA (H) treatment groups significantly increased hepatocyte survival and inhibited hepatocyte injury. In addition, compared with the APAP model group, the nanomedicine treatment could significantly reduce the contents of ALT and AST in the culture medium, as shown in Figure 7. In the high-dose group, compared to the free COSM, the nano-drug COSM@MSN-NH_2_-GA showed a more significant decrease in the indexes of ALT and AST. It can be speculated that the glycyrrhetinic acid receptor (GAR), as the most commonly used targeting effect, is overexpressed in hepatocytes [47,48]. MSN-NH_2_-GA uses a receptor-mediated strategy to improve delivery efficiency and achieve better therapeutic effect. Taken together, the results indicated that COSM@MSN-NH_2_-GA could treat APAP-induced hepatocyte injury. Therefore, the combined use of COSM and MSN-NH_2_-GA can improve the vitality of hepatocytes and provide great hope for the treatment of APAP-induced hepatocyte injury.

In this experiment, fluorescence microscopy was used to observe the cellular uptake behaviour of nanoparticles. As shown in Figure 8, at 5 min, no fluorescence could be observed in the C_6_-NP group, but blue nuclei stained with DAPI could be clearly observed. After 1 h, fluorescence was observed in the cytoplasm of the cells. The fluorescence intensity of the C_6_-NP group was significantly enhanced, indicating that the targeted nanoparticles showed higher cell uptake ability. At 2 h, the fluorescence intensity of the C_6_-NP groups confirmed an increasing trend. In the C_6_-NP group, the Merge diagram showed mixed blue and green light, and the cytoplasm and nucleus of the cells exhibited much fluorescence. This result may be due to the above-mentioned specific GA-receptor-mediated endocytosis mechanism, and a large number of C6-NPs nanoparticles are internalized by LO2 cells [29,30].

In this experiment, flow cytometry was performed to analyse the cell uptake behaviour of nanoparticles (Figure 9). At 5 min, the fluorescence intensity of the free coumarin group (938.6 ± 115.7) was not considerably longer than that of the C_6_-NP group (923.8 ± 27.4). At 1 h, the cell fluorescence intensity of the nanoparticle group was maintained at a high level (11378.07 ± 2692.05), which was appreciably different from that of the free drug group (3082.83 ± 220.94). With increases in the reaction time, the fluorescence intensity of the nanoparticle group reached 12991.77 ± 2303.26 at 2 h, whereas the fluorescence intensity of the free drug group was 4746.3 ± 1990.77. This finding shows that compared with free drugs, targeted modified nanoparticles exhibit a stronger drug transport ability and are more easily taken up by cells.

Overall, the uptake of glycyrrhetinic-acid-modified nanomedicines by cells was better. The results of fluorescence microscopy and flow cytometry showed that the glycyrrhetinic-acid-modified nanoparticles exhibited a higher fluorescence intensity than that of the free C_6_ group at the same time and showed stronger fluorescence intensity faster, indicating that the treatment can significantly increase the uptake of nanomedicines by LO2 cells.

## 3. Materials and Methods

### 3.1. Synthesis of MSN-NH_2_-GA Nanoparticles

The following substances were purchased from industrial suppliers and used as received. Triethanolamine (TEA), N-hydroxysuccinimide (NHS), dicyclohexylcarbodiimide (DCC), glycyrrhetinic acid (GA), dimethyl sulfoxide (DMSO), and diethyl ether were all biochemically obtained from MACKLIN Technology Co., Ltd. (Shanghai, China). Cyclohexane, absolute ethanol, and methanol were all provided by Tianjin Damao Reagent Factory (Tianjin, China). Cetyltrimethylammonium chloride (CTAC), tetraethyl orthosilicate (TEOS), 3-aminopropyltriethoxysilane (APTES), and 1-octadecene (ODE) were purchased from Sigma-Aldrich Trading Co., Ltd. (Shanghai, China). Unless otherwise indicated, all solvents were of analytical grade and were used without purification.

24 mL (25 wt %) CTAC solution and 0.09 g TEA were added to 36 mL distilled water and stirred at 150 rpm for 1 h. Then, 20 mL of ODE solution containing TEOS (10 *v*/*v* %) was added and reacted at 70 °C for 24 h to obtain MSN. MSN-NH_2_ was obtained by adding MSN to 25 mL anhydrous ethanol, adding 2.0 mL APTES, and reacting at 30 °C for 24 h. GA-NHS was obtained by adding 0.52 g NHS, 0.94 g DCC, and 1.0 g GA into 20 mL DMSO, and reacting at 40 °C for 24 h. Finally, GA-NHS and MSN-NH_2_ were added to DMSO and stirred at 60 °C for 24 h. MSN-NH_2_-GA was obtained by washing with DMSO and anhydrous ethanol for 3 times. Please see attachment Appendix A for details.

### 3.2. Preparation Characterization

FTIR spectra were obtained using an FTIR spectrophotometer (VERTEX 70v, Bruker GmbH, Bremen, Germany) to determine the profitable synthesis of MSN-NH_2_ and MSN-GA. The dimension distribution and zeta potential of the pattern nanoparticles were explored using a laser particle dimension analyser (Delsa, Beckman Technology Co., Ltd., Durahm, NC, USA). The morphologies of MSN and MSN-NH_2_-GA were observed by TEM (Tecnai G2 F20, Thermo Fisher Scientific, Waltham, MA, USA) and scanning electron microscopy (SEM) (XFlash 6130, Carl Zeiss, Oberkochen, Germany). The unique structures and pore dimension distribution traits of MSN and MSN-NH_2_-GA were determined by nitrogen adsorption (ASAP2460, American Mack Instruments Co., Ltd., Colonial Heights, VA, USA).

### 3.3. Encapsulation of COSM in MSN-NH_2_-GA (COSM@MSN-NH_2_-GA)

Chitosan oligosaccharide (COSM) was purchased from Shandong Aokang Biotechnology Co., Ltd., with batch number of 200409C, degree of deacetylation (DD) of 90.2%, and molecular weight of 1000 Da. To study the loading ability of COSM on the prepared nanoparticles, 10 mg of nanocarrier MSN-NH_2_-GA was placed into 10 mL of ethanol solution at room temperature and ultrasonically dispersed for approximately 10 min. Then, 10 mg of COSM was added and stirred slowly on a magnetic stirrer for 12 h. Finally, a high-speed centrifuge was used for centrifugation to collect the lower drug-loaded nanoparticles, which were freeze-dried for storage. Differential scanning calorimetry (DSC) curves were obtained using a synchronous thermal analyser (STA449, Netzsch, Selb, Germany) with a temperature range of 20 °C to 300 °C and a heating rate of 10 °C/min. The determination method of COSM content was established by DNS method, and the absorbance of COSM was determined by multifunctional enzyme-labeled instrument (MAXM4, Meigu Molecular Instruments Co., Ltd., Shanghai, China), and the standard curve of COSM was drawn (Appendix A). The amounts of COSM in the nanoparticles were measured.

The encapsulation efficiency (*EE%*) of COSM in the nanoparticles was calculated using the following formula:$$EE\;(\%)=\frac{Total\;amount\;of\;COSM\;drugs-COSM\;content\;in\;supernatant}{Total\;amount\;of\;nanoparticles}$$

The load capacity (*LC%*) of COSM in the nanoparticles was calculated using the following formula:$$LC\;(\%)=\frac{Total\;amount\;of\;COSM\;drugs-COSM\;content\;in\;supernatant}{Total\;amount\;of\;drugs\;administered\;by\;COSM}$$

### 3.4. Effect of COSM@MSN-NH_2_-GA on Cell Activity and Treatment

The cell line was purchased from the Cell Bank of the Chinese Academy of Sciences and was cryopreserved at the Institute of Traditional Chinese Medicine, Guangdong Pharmaceutical University, with the human foetal hepatocyte LO2 mobile line. The cells were cultured in RPMI-1640 medium containing 10% foetal bovine serum and 1% FBS in a humidified incubator with a 5% carbon dioxide atmosphere. This protocol was reviewed and approved by the Institutional Review Board of Guangdong Pharmaceutical University.

First, LO2 cells were cultured and observed by microscope. Then, according to the previous experimental results (see attachment: Appendix A), the administration components were divided into high, medium, and low (H, M, L) COSM, (H, M, L) MSN-NH_2_-GA nanoparticles and (H, M, L) COSM@MSN-NH_2_-GA nanomedicine, with 8 replicate wells in each group. APAP was added at the modelling concentration for 12 h, the model group was treated with APAP only, the clean group was treated with an identical amount of medium, and the experimental group was treated with free drug COSM, nanoparticles MSN-NH_2_-GA, and the nanodrug COSM@MSN-NH_2_-GA and cultured for 12 h.

The cytopathological state was observed to evaluate the therapeutic effect. In this study, the optimal concentration and time of APAP for modeling were screened (see attachment: Appendix A), and the cell viability was measured by CCK-8 method as the evaluation index. First, the cell culture medium was collected and centrifuged, and the contents of AST (C010-3-1, Nanjing Jiancheng Biological Co., Ltd., Nanjing, China) and ALT (C009-3-1, Nanjing Jiancheng Biological Co., Ltd., Nanjing, China) were measured according to the kit instructions. The survival state of adherent cells was detected by the CCK-8 method.

### 3.5. In Vitro Cellular Uptake

#### 3.5.1. Fluorescence Microscopy (CLSM)

Coumarin-6 (C_6_) is a liposoluble dye with strong fluorescence. It is commonly used in cellular uptake studies with nanoformulations [49,50]. LO2 hepatocytes with cell viability meeting the experimental requirements were inoculated into a 24-well plate and incubated for more than 12 h until 80% of the cells adhered to the wall. The C_6_ solution and C_6_-NP nanosolution were diluted with serum-free medium so that both groups of C_6_ concentrations were 800 ng/mL and were washed 3 times with PBS. After culturing for 30 min, 1 h, and 2 h, the original medium was discarded, subjected to three 5 min washes with PBS, incubated with 4% paraformaldehyde for 15 min, and washed 4 times with PBS after fixation. In a dark environment, the anti-fluorescence-quenching suspension-containing DAPI was dropped on the glass slide, and then the cell slide was placed upside down on the glass slide to complete the suspension. A fluorescence microscope was used for observation and analysis.

#### 3.5.2. Flow Cytometry (FCM)

LO2 cells were seeded in a 6-well plate at 20 w cells/well and incubated for more than 12 h, and cell adherence was observed. When more than 80% of the cells adhered, the coumarin solution and coumarin-labelled C_6_-NPs were diluted with minimal medium so that the fluorescein concentration was 800 ng/mL. Time gradients were set as 5 min, 1 h, and 2 h. After incubation for corresponding time in each group, cells were collected by centrifugation. The cells were resuspended with 300 μL PBS and detected by flow cytometry.

## 4. Conclusions

In this study, a multifunctional drug delivery carrier based on glycyrrhetinic acid embedded in silica nanoparticles was successfully synthesized. Not only does MSN-NH_2_-GA show satisfactory loading ability, but also it enhances its biocompatibility. The successful synthesis of MSN-NH_2_-GA was verified by FTIR, SEM and zeta potential measurement. In vitro studies show that this kind of nontoxic nanoparticles can significantly enhance the uptake of cells. The COSM drug has a protective effect on liver cell injury induced by APAP. In particular, the delivery of COSM through MSN-NH_2_-GA can greatly improve the therapeutic effect on LO2 cells. In summary, COSM@MSN-NH_2_-GA is a non-toxic, stable and efficient nano-therapeutic drug for acute liver injury. The results of this study provide a potential nanodelivery platform for the targeted therapy of acute liver injury.

## Data Availability

Data of the present study are available in the article.

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
