# Peer review of "Synthesis of Chitosan Oligosaccharide-Loaded Glycyrrhetinic Acid Functionalized Mesoporous Silica Nanoparticles and In Vitro Verification of the Treatment of APAP-Induced Liver Injury"

_molecules, 2023, doi:10.3390/molecules28104147_

Round 1

Reviewer 1 Report

In this manuscript entitled "Design the Synthesis and Evaluation of Chitosan Oligosaccharide Loaded with Glycyrrhetinic Acid Functionalized Mesoporous Silica Nanoparticles", the authors have synthesized a system based on ordered mesoporous silica for the controlled release of chitosan oligosaccharide. The work may be adequate. The comments may be useful for the improvement of the manuscript. Minor revisions are needed to make the work acceptable.

1.      The title selected by the authors does not reflect the complete purpose of the study. Evaluation for what?

2.       The footnote of Figure 3 is wrong. Also, this Figure should be displayed in two plots, one for the isotherms’ results and another one for the pore diameter distribution.

3.      Through XRD analysis, authors also can calculate the particle size (Scherrer’s equation) and lattice strain. These data could enrich the discussion. Authors should consider using this technique.

4.      From DSC results, authors could also calculate the organic content of MSN.

5.      Line 260, why is the materials and methods section in this part of the manuscript? Authors should revise all the manuscript’s sections before sending to the journal.

6.      The authors should add some references that include a series of mesoporous silica materials also used as adsorbents and drug delivery systems, such as 10.1002/adfm.201505073 and 10.1016/j.micromeso.2017.11.009.

Minor editing of English language required

Reviewer 2 Report

The authors fabricated mesoporous nanoparticles for targeted drug delivery. 

However the title is very questionable. It says "Design the synthesis and evaluation of ...." I am not sure what properties are aimed for evaluation. The title should be corrected. 

The authors showed a poor display of writing. For figure 2, I am not sure why the authors did FTIR. The peak 1445 and 1545 does not make any sense to me as I cannot see any peak there. 

For figure 3, the caption is wrong. 

The authors should add antimicrobial properties of the fabricated nanoparticles as it is important data for drug delivery. They can check the following article (https://doi.org/10.1021/acsabm.2c00502) 

The abstract should be rewritten by including at least some data/findings of the research. 

English should be improved.

Reviewer 3 Report

Dear authors,

please find attached my suggestions

Best regards

Round 2

Reviewer 2 Report

Can be accepted as it is.